# Structure, mechanism and crystallographic fragment screening of the SARS-CoV-2 NSP13 helicase

Joseph A. Newman [1✉], Alice Douangamath [2], Setayesh Yadzani [3], Yuliana Yosaatmadja [1], Antony Aimon [2], José Brandão-Neto[2], Louise Dunnett[2], Tyler Gorrie-stone[2], Rachael Skyner [2], Daren Fearon [2], Matthieu Schapira [3,4], Frank von Delft[1,2,5] & Opher Gileadi[1]

There is currently a lack of effective drugs to treat people infected with SARS-CoV-2, the cause of the global COVID-19 pandemic. The SARS-CoV-2 Non-structural protein 13 (NSP13) has been identified as a target for anti-virals due to its high sequence conservation and essential role in viral replication. Structural analysis reveals two "druggable" pockets on NSP13 that are among the most conserved sites in the entire SARS-CoV-2 proteome. Here we present crystal structures of SARS-CoV-2 NSP13 solved in the APO form and in the presence of both phosphate and a non-hydrolysable ATP analog. Comparisons of these structures reveal details of conformational changes that provide insights into the helicase mechanism and possible modes of inhibition. To identify starting points for drug development we have performed a crystallographic fragment screen against NSP13. The screen reveals 65 fragment hits across 52 datasets opening the way to structure guided development of novel antiviral agents.

[1] Centre for Medicines Discovery, University of Oxford, Oxford, UK. [2] Diamond Light Source Ltd., Harwell Science and Innovation Campus, Didcot, UK. [3] Structural Genomics Consortium, University of Toronto, Toronto, ON, Canada. [4] Department of Pharmacology and Toxicology, University of Toronto, Toronto, ON, Canada. [5] Faculty of Science, University of Johannesburg, Johannesburg, South Africa. ✉email: Joseph.Newman@cmd.ox.ac.uk

SARS-CoV-2 is the causative agent of the current global coronavirus (COVID-19) pandemic, a severe respiratory disease that emerged in the Chinese city of Wuhan in late 2019[1,2]. Genome sequencing indicates a zoonotic origin for SARS-CoV-2[3], as was the case for previous coronavirus outbreaks of SARS in 2002[4] and MERS in 2012[5]. SARS-CoV-2 belongs to the genus *Betacoronavirus* which have a positive sense single stranded RNA genome of approximately 30 KB in length and amongst the largest know of any viral RNA genomes. The SARS-CoV-2 genome encodes two open reading frames ORF1a and 1b, that when translated produce polyproteins that are processed by viral proteases into 16 non-structural proteins (NSP1-16)[6] that collectively form the machinery for viral replication and transcription.

NSP13 is a 67 kDa protein that belongs to the helicase superfamily 1B, it utilizes the energy of nucleotide triphosphate hydrolysis to catalyze the unwinding of double-stranded DNA or RNA in a 5′ to 3′ direction[7]. Although NSP13 is believed to act on RNA in vivo enzymatic characterization shows a significantly more robust activity on DNA in in vitro assays with relatively weak non processive helicase activity when compared to other superfamily 1B enzymes[8,9]. NSP13 has been shown to interact with the viral RNA-dependent RNA polymerase NSP12[10,11], and acts in concert with the replication-transcription complex (NSP7/NSP8/NSP12)[12]. This interaction has been found to significantly stimulate the helicase activity of NSP13 possibly by means of mechano-regulation[11,13]. In addition to its helicase activity, NSP13 also possesses RNA 5′ triphosphatase activity within the same active site[14], suggesting a further essential role for NSP13 in the formation of the viral 5′ mRNA cap.

NSP13 contains 5 domains, a N-terminal Zinc binding domain (ZBD) that coordinates 3 structural Zinc ions, a helical "stalk" domain, a beta-barrel 1B domain and two "RecA like" helicase subdomains 1 A and 2 A that contain the residues responsible for nucleotide binding and hydrolysis. This same basic 5-domain architecture is shared by by other Nidovirus helicases such as the NSP10 proteins from Equine arteritis virus[15] and Porcine reproductive and respiratory syndrome virus[16] and to a lesser extent the human nonsense-mediated mRNA decay factor UPF1[17], which feature a structurally similar helicase core (R.M.S.D around 3.0 Å) and more diversity in terms of composition of the zinc-binding domain and connection between the 1B domain and helicase core. Previous X-ray structures of NSP13 have been solved for MERS-CoV and the highly related SARS-CoV to 3.0 Å and 2.8 Å respectively[10,18]. More recently, cryo-electron microscopy studies have revealed the architecture of the NSP13 containing replication and transcription complex, which contains two copies of NSP13 that interact with NSP8 via the N-terminal ZBD[12,19,20]. One of the NSP13 protomers makes additional interactions with NSP12 and is located with its RNA binding site in the path of downstream RNA[12]. However, the polarity of the helicase translocation 5′ to 3′ and polymerase are in opposition, leading to the suggestion that NSP13 may play a role in backtracking, template switching or disruption of downstream secondary structures[12]. It is also not well understood why two copies of the helicase are present, although mutagenesis of specific residues involved in discrete domain contacts indicate a role for both domains for the enhanced helicase activity of the complex[19].

Whilst the precise role for NSP13 in the viral life cycle has yet to be determined, it was found to be a critical component for viral replication in the highly similar SARS-CoV[10], and in other viral species more distantly related to SARS-CoV-2[21–23]. For this reason, NSP13 has been suggested as a good target for the development of new antiviral drugs[24,25]. To this end, several efforts have identified compounds which inhibit SARS-CoV NSP13 with IC50's in the low µM range and display anti-viral activity in cellular assays[26–28]. The lack of structural information on the binding mode and unknown mode of action of these compounds is likely a barrier to further development although these studies demonstrate that NSP13 may be a tractable target. NSP13 is also among the most conserved of the non-structural proteins in the SARS-CoV-2 genome, differing from SARS-CoV in only a single amino acid (V570I). Thus, compounds targeting SARS-CoV-2 NSP13 would likely be effective against SARS-CoV and potentially other future emerging coronaviruses, making it an ideal target for the development of new antiviral therapeutics with a broad spectrum of action.

To accelerate the development of anti-viral therapeutics targeting NSP13 we have determined the X-ray crystal structure to high resolution in a variety of nucleotide bound and conformational states. These structures provide insights into the catalytic mechanism and provide a robust platform for virtual screening. We have analyzed potential druggable pockets on NSP13 and find two such pockets, one of which is highly conserved across a variety of viral species. Finally, we have performed a crystallographic fragment screen on NSP13 which demonstrates the ligandability of these pockets and provides structural information and chemical starting points for the rational design or structure-guided optimization of new anti-viral therapeutics.

## Results

**Crystal structures of NSP13 in APO, phosphate and nucleotide bound form.** We have determined the crystal structures of full-length SARS-CoV-2 NSP13 (residues 1-601) in three different forms that represent different conformations of the ATPase catalytic cycle. The APO form was determined at 2.4 Å resolution with two NSP13 molecules in the asymmetric unit. A phosphate-bound form was determined with the same crystal form but diffracted to significantly higher resolution (1.9 Å). A nucleotide bound form of NSP13 with the ATP analog AMP-PNP was determined to 3.0 Å resolution in a monoclinic crystal system with 4 molecules of NSP13 in the asymmetric unit (Supplementary Fig. 1). In general, the electron density is of high quality throughout (Supplementary Fig. 2), with the exception of the final 7 residues at the C-terminus and two loops within the 1B domain (residues 185–194 and 203–207), which display variable degrees of disorder across the various chains. The models have been restrained to standard bond lengths and angles and a summary of the data collection and refinement statistics can be found in Table 1.

All chains feature the same basic architecture which has been described previously for NSP13 structures of related organisms[10,18] (Fig. 1a). The ZBD coordinates two structural zinc ions via a RING like module in the N-terminus and an additional single zinc ion via a treble-clef Zinc-finger. The ZBD packs against the three-helix bundle of the "stalk" domain which also makes contacts with 1A domain on its opposite side. Following closely from the end of the third helix of the stalk domain is the 1B domain which forms a 6-stranded RIFT type anti-parallel β-barrel[29]. A long 30 amino acid linker that does not adopt any secondary structure links the 1B to the 1A and 2A RecA-like core helicase domains. The nucleotide binding site is situated in a cleft between the 1A and 2A domains with specific contacts to the nucleotide provided by conserved helicase motifs I, II, and III in the 1A domain and IV, V, and VI in the 2A domain.

**Comparisons of NSP13 nucleotide bound structures in different conformational states.** In the nucleotide bound form, AMP-PNP is bound to all 4 molecules in the asymmetric unit but in two distinctly different modes (Fig. 1b, c). In mode A (exemplified by chain A of the nucleotide form) both AMP-PNP and a hydrated $Mg^{2+}$ ion are bound, with the adenine base making stacking interactions to H290 on one side and R442 on the other.

**Table 1 Data collection and refinement statistics.**

|  | Phosphate bound | APO | AMP-PNP |
|---|---|---|---|
| Space group | P1 | P1 | C 2 |
| Cell dimensions, $a$, $b$, $c$ (Å) | 59.1, 70.1, 84.6 | 56.7, 70.1, 84.0 | 324.9, 59.5, 132.4 |
| Angles $\alpha$, $\beta$, $\gamma$ (°) | 102.5, 95.6, 112.8 | 104.4, 93.3, 112.2 | 90, 93.9, 90 |
| Wavelength (Å) | 0.91 | 0.91 | 0.91 |
| Resolution (Å) | 62.1–1.94 (1.94–1.92) | 80.2–2.20 (2.27–2.20) | 81.0–3.04 (3.09–3.04) |
| $R_{merge}$ | 0.053 (0.904) | 0.077 (0.459) | 0.258 (1.390) |
| $I/\sigma I$ | 11.1 (1.1) | 6.1 (0.7) | 6.1 (1.2) |
| CC1/2 | 0.998 (0.669) | 0.946 (0.605) | 0.978 (0.359) |
| Completeness (%) | 97.3 (97.2) | 91.8 (90.7) | 97.7 (95.2) |
| Multiplicity | 3.5 (3.2) | 2.5 (2.5) | 3.4 (3.2) |
| No. Unique reflections | 86341 (4418) | 53358 (4635) | 48372 (2343) |
| *Refinement statistics* | | | |
| Resolution | 57.6–1.94 | 80.2–2.20 | 81.0–3.04 |
| $R_{work}/R_{free}$ (%) | 20.9/25.3 | 22.9/28.6 | 24.4/28.4 |
| No. atoms | | | |
| Protein | 8917 | 9042 | 18063 |
| Solvent | 456 | 221 | 3 |
| Ligand/ion | 26 | 6 | 137 |
| Average B factors (Å2) | | | |
| All atoms | 53 | 46 | 62 |
| Protein | 54 | 46 | 62 |
| Solvent | 51 | 42 | 35 |
| Ligand/ion | 50 | 41 | 77 |
| Wilson B | 38 | 32 | 50 |
| RMS deviations | | | |
| Bond lengths (Å) | 0.005 | 0.003 | 0.004 |
| Bond angles (°) | 0.75 | 0.57 | 0.85 |
| Ramachandran plot | | | |
| Favored (%) | 96 | 94 | 95 |
| Allowed (%) | 4 | 6 | 5 |
| PDB ID | 6ZSL | 7NIO | 7NN0 |

The ribose O3 makes a single hydrogen bond to E540 (part of motif V), and the α-phosphate interacts with the main chain of H290 and the side chain of K320 via a salt bridge. The β-phosphate is situated directly within the phosphate binding motif I and makes interactions with the main chain amides of residues 287–289. The γ-phosphate makes extensive interactions with the sidechains of Q404, R443 and R567 (from motif III, IV, and VI respectively). The $Mg^{2+}$ ion is in an octahedral coordination and makes direct contacts to both the β and γ phosphates as well as the side chain of S289 and additional three waters. The conserved D374 and E375 from motif II (walker B) make further interactions with the waters and are poised to perform their presumed catalytic role of activating a hydroxyl for nucleophilic attack (Fig. 1b).

In nucleotide binding mode B (chains B, C and D of the nucleotide form) the interaction environment of the adenine base and γ-phosphate are largely unchanged, with the ribose O2 contacting the side chain of K320. The α-phosphate is situated in the equivalent position of the β-phosphate from mode A, with the β-phosphate interacting with R443 and the invariant K288 from motif I contacting oxygens on both α and γ phosphates (Fig. 1c). No density could be observed for $Mg^{2+}$ ions in these three chains with the sidechain of S289 contacting instead the α-phosphate and the residues from motif II more distant. In the phosphate bound structure the two phosphates bind in essentially equivalent positions to the α and γ phosphates from binding mode B, although very slight shifts are observed for the equivalent of the γ-phosphate which probably represents the expected binding poise for the hydrolyzed phosphate product (Supplementary Fig. 3).

In our structures, the majority of the contacts made around the adenine and ribose moieties do not appear to be specific to a single nucleotide, consistent with the observation that NSP13 is able to hydrolyze multiple nucleotides including deoxy-ribonucleotides in

enzymatic assays[14,30]. A single contact between the adenine N6 to the main chain carbonyl of E261, and the ribose 2′ OH and K320 indicates a possible preference for both adenine and ribonucleotides, consistent with preferences identified in recent single molecule analysis of NSP13 in vitro[30]. The structures of NSP13 in complex with nucleotide also offer insights into the ability of NSP13 to perform the RNA 5′ triphosphatase reaction. In both binding modes (but particularly mode A) large portions of the adenine and ribose moiety are solvent exposed with the ribose O3 group pointing towards the outside of the active site such that it would be possible to accommodate additional phosphodiester linkages such as those found on a 5′ triphosphate containing RNA without steric clashes (Supplementary Fig. 4).

**Flexibility and domain movements in the structures of NSP13.**
In addition to the specific changes in the nucleotide binding site detailed above there are also significant domain movements that become evident when comparing the various chains of NSP13 structures. A systematic pairwise comparison of Cα RMSD values for all NSP13 chains from this study and relevant PDB entries is shown in Supplementary Table S1. Whilst the individual domains of NSP13 are generally well conserved (in the region of 0.8 Å RMSD or lower), the differences when comparing entire chains are in some cases significantly greater (in the region of 3 Å). These differences can be attributed to rigid domain movements and structural superpositions identify multiple sources of domain movements (Fig. 2a–d). Firstly, the Zinc binding domain appears to be fairly flexible with rotations of 10–15° around a single hinge point located near residues 104–105 (Fig. 2c). There is no obvious connection between the conformation of this domain and crystal form or nucleotide bound status as variability between chains within a single crystal forms is equally large as when comparing

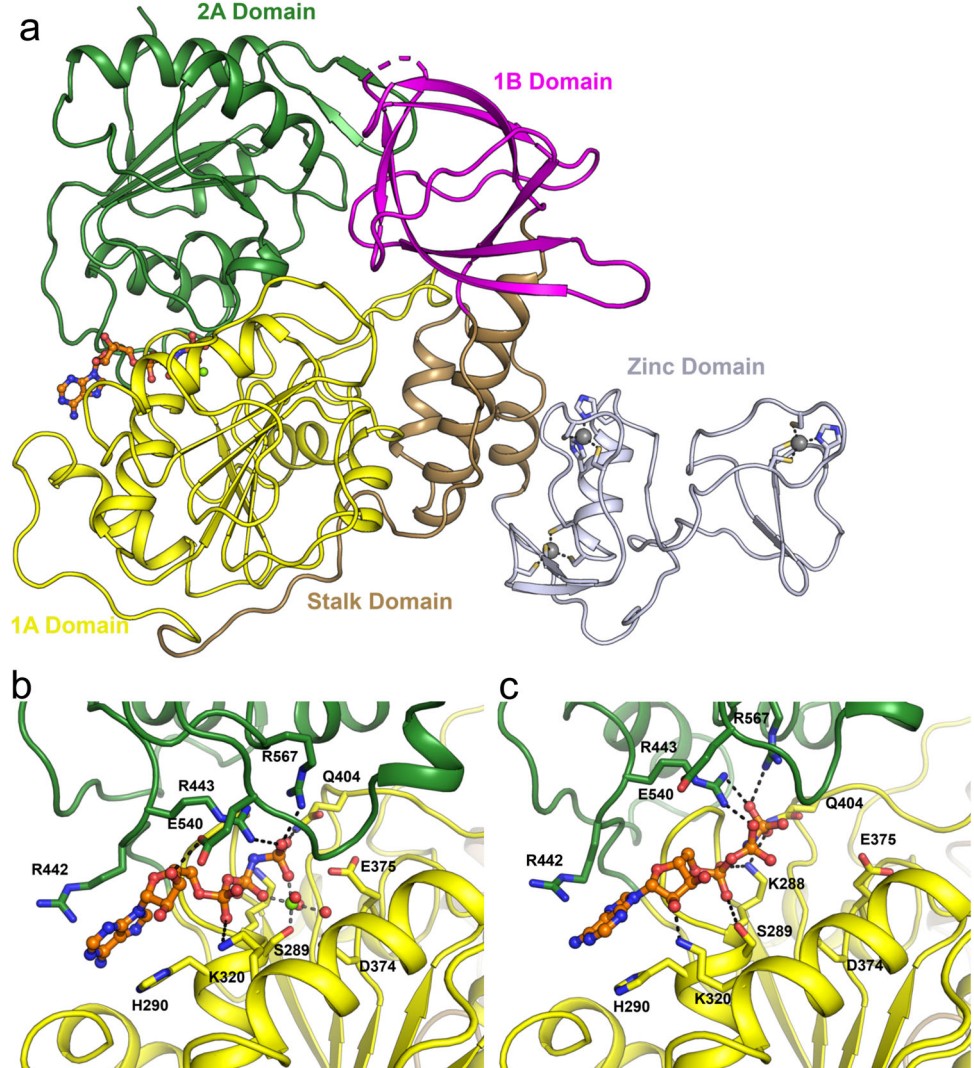

**Fig. 1 Overall structure of SARS-CoV-2 NSP13. a** Structure overview with domains labeled and colored individually (the same color scheme is used throughout). The AMP-PNP nucleotide is shown in stick format in the nucleotide binding site between the 1 A and 2 A domains. **b** Close up view of the nucleotide binding mode for the AMP-PNP Mg$^{2+}$ complex (mode A), with interacting residues labeled and shown in the stick format. **c** Close up view of the nucleotide binding form the AMP-PNP complex (mode B), viewed from the same orientation as panel (**b**).

between forms. More directly related to the nucleotide status is the relative conformations adopted by the 1A and 2A domains which shifts by around 20° and adopts a closed conformation the Mg$^{2+}$ AMP-PNP bound crystals structure, and a more open conformation in the APO, Phosphate bound and other AMP-PNP bound chains (Fig. 2d). The 1B domain also displays significant conformational variation with rotations of around 25° (Fig. 2b). In this case the movements do correlate with the relative positioning of the 1A and 2A domains, and the 1B domain can be seen to pivot around its narrow point of connection to the stalk domain. In both the open and closed states, the 1B domain forms the same basic interface to the helicase core, forming contacts to loop 337–340 in the 1A domain and loops 483–487 and 513–517 in the 2A domain (Fig. 2b).

**Model for the translocation mechanism of NSP13.** Previous structural studies on UPF-1[17] and recent Cryo-EM structures of NSP13[19] within the viral replication-transcription complex reveal the expected DNA/RNA binding site which is formed from a channel bounded on one side by the 1A and 2A domains and the

other by the 1B and Stalk domains. Although the precise details are still to be revealed, the RNA at its 5′ end appears to interact with the 2A domain via the phosphodiester backbone with the nucleobases pointing towards the 1A and 1B domains, whilst at the 3′ end the RNA backbone contacts the 1A domain with the bases pointing towards the stalk and 1B domain (Fig. 3a). Positioning RNA into its expected binding site in both the more "open" APO/product state and the "closed" ATP-Mg$^{2+}$ bound conformations, the RNA contacting motifs on the 1A and 2A domains are shifted such that conserved contacts may be formed to phosphates that differ by a single RNA base. This observation has allowed us to construct an inchworm type mechanistic model for the translocation of NSP13 along single stranded RNA in a 5′ to 3′ direction. In our model the "closed" conformation is assumed to represent an activated pre-hydrolysis state which interacts with the RNA most strongly via the 2A domain and more weakly via the 1A domain at the 3′ end (Fig. 3a). ATP hydrolysis triggers the conformational change to the open product state with the 1A domain sliding a single nucleotide step toward the 3′ end of the RNA (Fig. 3a). Based on the requirement for directional movement, we assume that following this

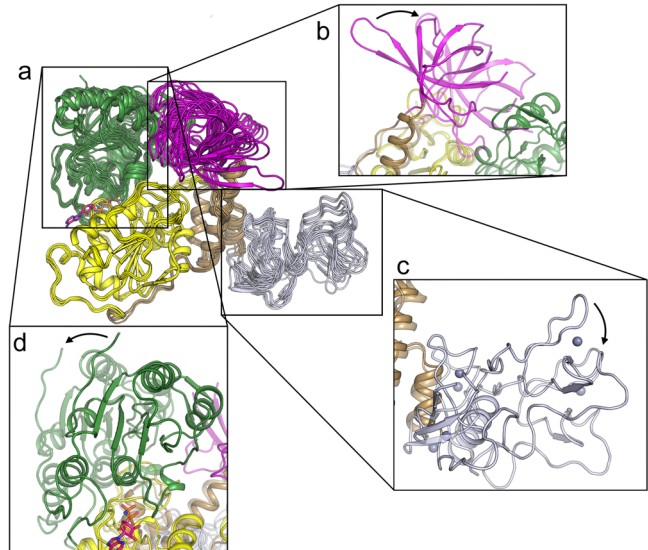

**Fig. 2 Conformational flexibility exhibited in the NSP13 structures.**
**a** Overall view showing a structural superposition of the phosphate bound (2 chains), APO (2 chains) and AMP-PNP bound (4 chains) NSP13 structures. The 1A domain was used as the reference for the structural superposition. **b** Close up view of the variable conformation of the 1B domain, the two most different conformations are shown. **c** Close-up view of the variable conformation of the Zinc domain. **d** Close-up view of the variable conformation of the 2A domain.

structural transition, the 1A domain would switch to bind the RNA more tightly either as a result of the remodeled RNA interface or from ADP and phosphate release. The nature of the nucleotide binding site would suggest that ADP would be released before phosphate due to steric effects. The catalytic cycle is completed by binding a new ATP molecule which triggers the closed conformation, although this time the 2A domain moves relative to the RNA (Fig. 3a and Supplementary Movie 1). A requirement of this model is a mechanism for ATP binding and subsequent hydrolysis to induce conformational changes and for the RNA binding affinity of one or both of the 1A or 2A domains to be modulated according to the nucleotide or conformational status. Comparing the mode of nucleotide binding in the open and closed states, the nucleotide is bound in two different modes, with the γ-phosphate in mode B (product state) being considerably more distant from the phosphate binding residues in motif I. We presume that charge repulsion following ATP hydrolysis would induce this conformation and movement of the 2A domain is induced due to the extensive interactions between it and the γ-phosphate provided by residues R443, Q537, and R567 from motifs IV, V, and VI respectively (Fig. 3b). The means by which the RNA binding affinities may be modulated are more uncertain partly due to the lack of a detailed understanding of the RNA-protein interface in both conformational states. One possibility is that this remodeling is a result of the coordinated movements of the 1B domain which forms a significant part of the RNA protein interface and appears to contact the RNA more extensively close to the 1A domain in the open conformation and close to the 2A domain in the closed conformation (Fig. 3c). The calculated interface areas between RNA and protein is thus more extensive with the 3′ end in the closed conformation (558 Å$^2$ versus 528 Å$^2$) and with the 5′ end in the open conformation (595 Å$^2$ versus 574 Å$^2$), matching the preference required for 5′ to 3′ directional translocation. Another possibility is that the RNA interface is remodeled via contacts from regions proximal to key helicase motifs such as the loop following motif III in the 1A

domain, and helices preceding motif V and following motif VI in the 2A domain which form part of the RNA interface whilst also contacting the γ-phosphate or hydrolyzed product in a conformation-dependent manner (Fig. 3b).

This mechanistic model is similar to the mechanisms suggested previously for other SF1B helicases such as RecD2[31], and provides a framework for the understanding of the NSP13 translocation mechanism and possible sites of inhibition including possible allosteric sites that may differ between the two states and block structural transitions that occur as part of the catalytic cycle. We do not describe an active base separating mechanism for NSP13 consistent with recent biochemical and single molecule studies of NSP13. In the RNA unwinding reaction, NSP13 was a predominantly passive helicase (advancing upon the spontaneous opening of base pairs), with a peak step size of 2 base pairs (interpreted to be 2 rate-limiting ATP binding events) and a strong force-dependent stimulation of activity that suggests mechanoragulation by the RNA polymerase NSP12[8]. Single molecule FRET studies of DNA unwinding by NSP13 show larger step sizes of up to 4–9 base pairs depending on nucleotide used which the authors suggest may indicate a "spring loaded" unwinding mechanism with the flexible 1B and Stalk domains[30]. Whilst the larger step sizes of the unwinding mechanisms contrast with our single step translocation mechanism, all models feature the same stoichiometry of a single nucleotide hydrolysis event per base. We suggest that the lack of an active strand separating hairpin or wedge to aid strand separation NSP13 is unable to translocate with the same efficiency and must presumably pause to either accumulate tension or wait for the spontaneous opening of DNA/RNA.

**X-ray crystallographic fragment screening of NSP13.** To identify possible starting points for the development of NSP13 inhibitors we have performed an X-ray crystallographic fragment screen. The phosphate-bound NSP13 crystals show robust crystallization behavior and routinely diffract to around 2.0 Å resolution, sufficient for the reliable identification of binders. 648 crystals were soaked individually with a library of chemical fragments at a ~50 mM final concentration. X-ray analysis of the majority of crystals showed no bound compounds; however, 65 bound fragments were found within 52 datasets using the PANDDA algorithm[32]. Some of the hits were found in pockets predicted to be of functional importance, including the nucleotide and nucleic acid binding sites (Fig. 4a and Supplementary Table I; the bound fragment structures are be referred to by the corresponding PBD depositions). In the nucleotide binding site, 15 fragments were bound in positions overlapping with the ATP ribose and adenine moiety (Fig. 4b). three of these fragments (PDB:5RLI, 5RLJ, and 5RLW) contain sulfonamide functional groups which make polar contacts to key residues within motif I and the phosphate ion occupying the α-phosphate position. The rest of the fragments explore the wider vicinity of the pocket occupied by the adenine moiety, making interactions with nearby residues such as H290, K320, Y342, R442, and N464.

Several fragment clusters were found in or close to the RNA/DNA binding channel (Fig. 4c). Close to the 5′ end of the channel, three fragments (5RLH, 5RLZ, and 5RMM) occupy a pocket formed between the 2A and 1B domains and make polar contacts to residues S486, N516, Y515, and T552 on the 2A domain (Fig. 4c and S5). These contacts are direct mimics of contacts formed by two successive RNA phosphates in the structure of the related UPF1 helicase in complex with RNA[17], making these fragments particularly attractive starting points for the design of RNA competitive inhibitors. In the central cavity of the channel a single fragment (5RML) occupies a hydrophobic

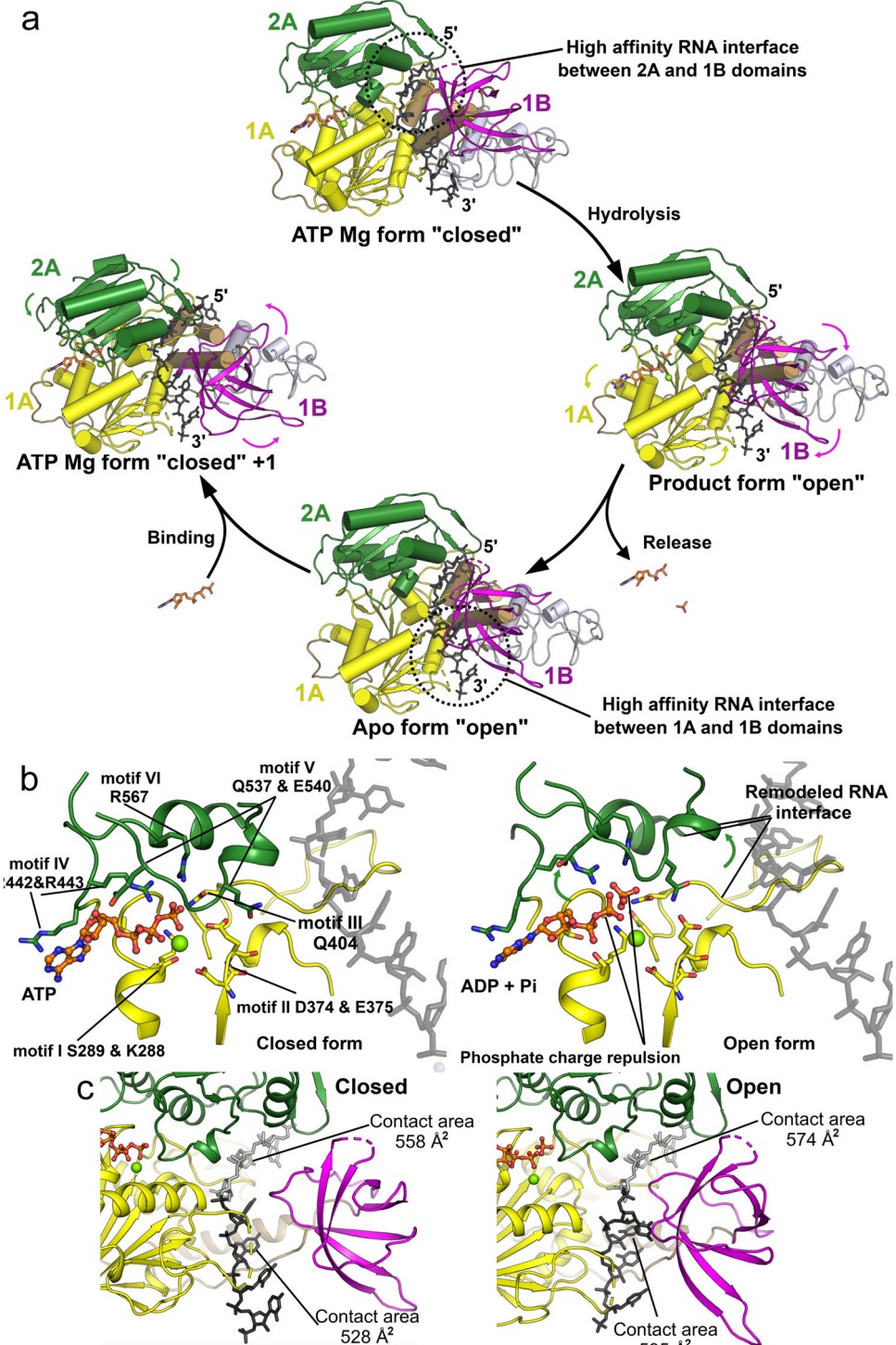

**Fig. 3 Model for the NSP13 5′ to 3′ translocation mechanism. a** Proposed translocation mechanism for NSP13 based on the transition from the closed (pre hydrolysis) to open (Product and APO) forms. The transitions are initiated by the binding, hydrolysis and release of ATP which triggers the conformational changes and remodels the RNA interface. **b** Close up view of the active site with ATP in the closed conformation (left) and ADP and Pi in the open conformation (right). Hydrolysis as subsequent charge repulsion could trigger the opening of the cleft between the two domains with conserved motifs on the 2A domain primarily contacting the product phosphate whilst the ADP product interacts with the 1A domain. Several of the phosphate interacting motifs are proximal to regions of the RNA binding interface indicating the possibility of modulation based on hydrolysis status. **c** Remodeling of the RNA interface based on the position adopted by the 1B domain. The closed conformation shows on the left and the open on the right. The contact areas for the 5′ and 3′ RNA regions (depicted in gray and black) is shown.

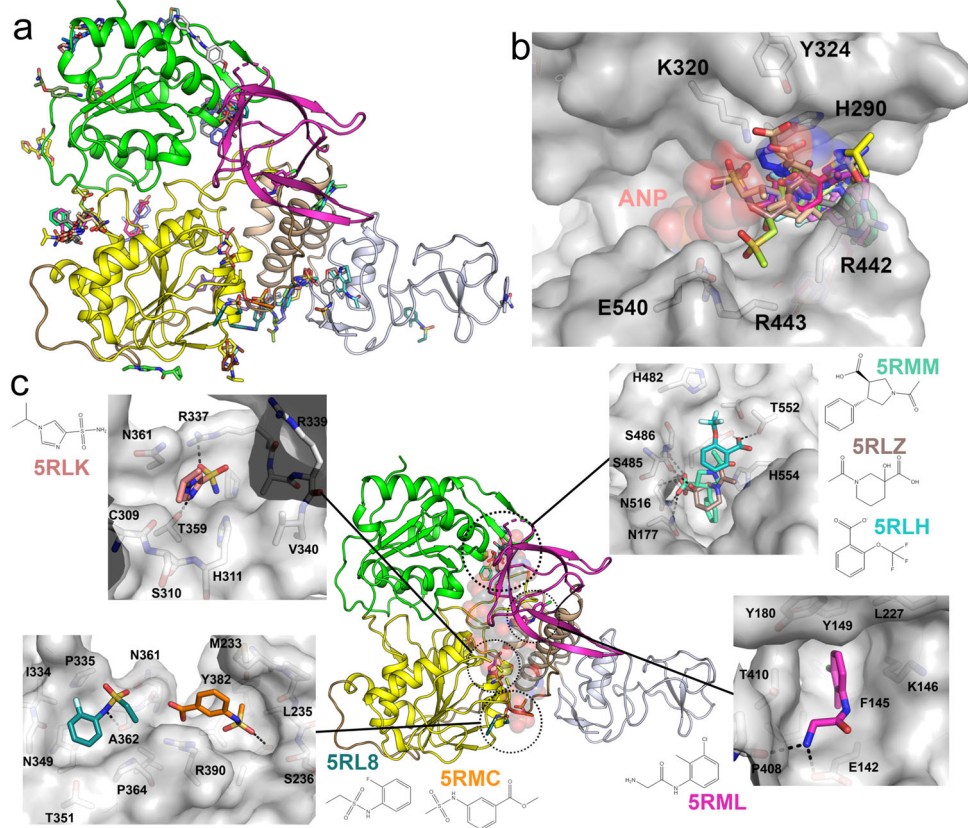

**Fig. 4 Fragments bound to NSP13 in the nucleotide and RNA binding interfaces. a** Overall view of all fragments bound to NSP13. **b** Close-up view of the 15 fragments identified in the nucleotide binding site. The AMP-PNP moiety (ANP) is shown in a semi-transparent sphere representation for reference. **c** Overview of the fragments bound to the NSP13 RNA interface. The main panel shows the positioning of the clusters with the RNA shown in a semi-transparent sphere representation for reference. Each binding interface is shown in the inset in a detailed surface view with polar contacts shown in black.

pocket created by the side chains of F145, K146, Y149, Y180, and T410, and makes further polar interactions to E142 and P408 on the periphery (Fig. 4c). A single sulfonamide containing fragment (PDB entry 5RLK) occupies a similar conserved phosphate binding region on the 1A domain and makes contacts to T359 and H311 as well as inducing the ordering of loop 337–340 (via contacts to R337) which is disordered in the ground state model (Fig. 4c and Supplementary Fig. 6). This fragment would also be expected to be RNA competitive based on the UPF1 RNA complex[17]. Also bound near the RNA interface where the 3′ end would be expected to exit are two fragments (5RL8 and 5RMC) which form contacts to residues in the 1A domain and would be expected to block RNA from entering the cleft.

Our analysis of the NSP13 mechanism and flexibility also suggests the importance of fragments that may be potential starting points for allosteric inhibitors, due to binding in sites that are specific to one conformational state and may block structural transitions that occur as part of the catalytic cycle. In addition to the nucleotide and RNA binding pockets identified above, three fragment sites were identified that span domains that exhibit conformational variability. The most prominent of such sites is found in a cleft between the Zinc domain and the stalk domain, which bound to 11 fragments, several of which share a common mode of interaction and are candidates for fragment merging (Fig. 5a). Two fragments (5RMF and 5RMB) were bound to a shallow, predominantly hydrophobic pocket between the 1A and 2A domains that is formed on the opposite face to the nucleotide-binding site, and appears to be present only in the open product state conformation (Fig. 5b). A further 3 fragments were bound to at the junction of the 1B and stalk domain (5RL6, 5RL7, and

5RLU) which is also close to the 3′ end of the RNA binding interface (Fig. 5c), although part of this site is formed by contributions from a crystallographic neighbor.

**Druggability and conservation analysis of NSP13 binding pockets**. In addition to the experimental fragment screening, we used a computational approach (ICM; Molsoft, San Diego) to find potentially druggable binding sites in our NSP13 structures. We identified two pockets of interest that are expected to be functionally relevant (Fig. 6a): the binding site occupied by AMP-PNP, at the interface of domains 1A and 2A, and a pocket lined by domains 1A, 1B, and 2A which is occupied by the 5′-end of the substrate RNA in the SARS-Cov-2 transcription complex [PDB:7CXM][19]. This 5′-RNA pocket is also occupied by some of our soaked fragments [PDB:5RMM, 5RLH, 5RLZ]. Both sites are accessible in the context of the transcription complex structure (Fig. 6a), suggesting that they could be targeted pharmacologically. We used SiteMap to evaluate the druggability of these two sites[33]. The 5′-RNA pocket is clearly druggable (druggability score 1.03). The nucleotide site is highly charged and developing drugs targeting this site is predicted to be more challenging (druggability score 0.91).

Ideally, a drug against COVID-19 would also be effective against past and future coronaviruses. To behave as a broad-spectrum inhibitor, a pharmacological agent targeting NSP13 would need to exploit a binding site that is highly conserved across coronaviruses. To evaluate the relevance of the nucleotide and 5′-RNA binding pockets for the development of pan-coronavirus drugs, we analyzed the conservation of amino-acids lining these two pockets across twenty-seven α- and β-

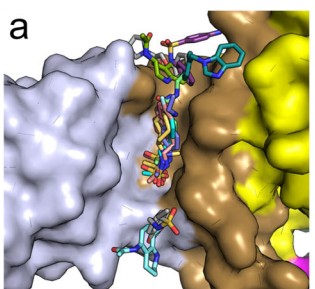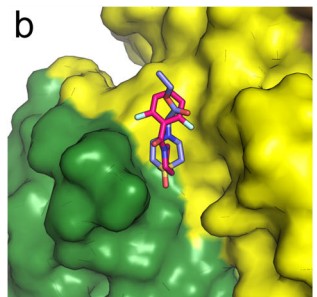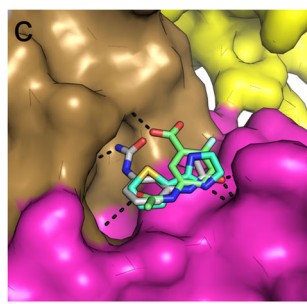

**Fig. 5 Fragments bound on the interfaces between domains that appear to move as part of the catalytic cycle and thus may be starting points for allosteric inhibitor design. a** Surface view of a prominent fragment binding site which bound to 11 fragments in a cleft between the Zinc and Stalk domains with the domains colored individually as for the color scheme in Fig. 1. **b** Two fragments were observed to bind in a cleft between the 1A and 2A domains approximately opposite the hinge. **c** Three fragments were bound in a shallow pocket between the Stalk and 1B domain (also close to the RNA interface) and make polar contacts to both regions.

coronaviruses with reviewed sequences in Uniprot. We find that 79% and 87% of residues lining the nucleotide and 5′-RNA sites respectively are conserved in all analyzed coronaviruses (Fig. 6b). In addition, 100% of side-chains in direct contact with our co-crystallized fragments (PDB:5RMM, 5RLZ) are conserved. In a systematic analysis of the sequence conservation of nineteen binding pockets from fifteen SARS-CoV-2 proteins across the same twenty-seven coronaviruses, we find that the active site of RdRp/NSP12 (94% sidechains conserved) and the ADP-bound pocket of the NSP12 NiRAN domain (87% sidechains conserved) are the only two cavities with a degree of conservation as high as the 5′RNA site of NSP13 (Yazdani et al. bioRxiv 2021). Together, our results indicate that the binding pocket occupied by the 5′-end of the RNA substrate is druggable and highly conserved. As such it is a good candidate for the development of broad-spectrum inhibitors.

## Discussion

We have determined the crystal structure of SARS-CoV-2 NSP13 in APO, phosphate-bound, and nucleotide bound states. These structures are of good quality and the crystals diffract to significantly higher resolution than previous structures of related NSP13 proteins and are thus a good starting point for virtual ligand screening. Our analysis of the structures of NSP13 revealed a high degree of conformational heterogeneity, with two distinct "open" and "closed" forms identified in the AMP-PNP complex crystals which represent different states in the catalytic cycle. We have used this information to suggest a translocation mechanism for NSP13 based on the transition between these states, with concomitant modulation of RNA binding interfaces producing directional movement along single stranded RNA. We have utilized the robust crystallization and high-resolution diffraction of the phosphate bound crystals to perform crystallographic fragment screening on NSP13. The screen identified over 50 binders with fragment hotspots in both nucleotide and RNA binding channels that can be used as starting points for design of novel anti-virals. Our analysis of the mechanism has also allowed us to identify fragments that may serve as possible starting points for allosteric inhibition based on their ability to interfere with or block structural transitions that form part of the catalytic cycle. Whilst these fragments do represent useful starting points for inhibitor development they have not yet been validated as inhibitors in biochemical assays or as binders in alternate biophysical assays and are not expected to be potent inhibitors without further optimization (see, for example,[34–36]). Finally, we have assessed the druggability and sequence conservation of pockets on NSP13, this analysis shows that a fragment containing pocket on the 5′ end of the RNA binding site is highly druggable and

amongst the most well-conserved pockets in the entire SARS-CoV-2 proteome, making it a good target for the development of anti-viral therapeutics that may be able to combat the current pandemic and also future emerging viral threats.

## Methods

**Cloning and expression of NSP13.** The plasmid for N-terminally His-ZB tagged NSP13 was synthesized in a pNIC-ZB vector (Twist biosciences) with codon optimization for expression in *Escherichia coli* (Supplementary Table 4). The plasmid and its full sequence have been deposited in Addgene (https://www. addgene.org/159614/). For overexpression, the plasmid was transformed into *E. coli* BL21 Rosetta2 cells. Cell cultures were grown in Terrific Broth media at 37 °C, with shaking at 180 rpm. When the $OD_{600}$ reached 2–3, IPTG (300uM) was added to the media and cultures were incubated overnight at 18 °C, shaking 180 rpm.

**Protein purification.** Cell pellets were re-suspended in lysis buffer (50 mM HEPES pH 7.5, 500 mM NaCl, 5% Glycerol, 10 mM Imidazole, 0.5 mM TCEP) with protease inhibitors (Merck Protease inhibitor cocktail III, 1:500). Cells were disrupted by sonication for 15 min 10 s on 5 s off, and cell debris was removed by centrifugation in a JA25.5 rotor at $72,400 \times g$ for 30 min. The supernatant was incubated for 40 min with 5 ml of Ni resin (IMAC sepharose) for batch binding. The tubes containing the lysate were centrifuged at $700 \times g$ at 4 °C for 5 min and the supernatant discarded. Beads were loaded on a gravity flow column and washed with 40 ml lysis buffer, 25 ml wash buffer (50 mM HEPES pH 7.5, 500 mM NaCl, 5% Glycerol, 45 mM Imidazole, 0.5 mM TCEP). A further wash with 10 ml Hi-salt buffer (50 mM HEPES pH 7.5, 1 M NaCl, 5% Glycerol, 0.5 mM TCEP) and again with another 10 ml of wash buffer. Proteins were eluted with addition of 15 ml of elution buffer (50 mM HEPES pH 7.5, 500 mM NaCl, 5% Glycerol, 300 mM Imidazole, 0.5 mM TCEP). The elution fraction was immediately applied to a 5 ml HItrap SP column using a syringe, collecting the flow through. The SP column was washed with 10 ml elution buffer and proteins were eluted with 15 ml Hi-salt buffer. The NSP13 protein was found to be present in flow-through and elution fractions and both fractions were pooled and treated separately from this point onward. For further purification protein samples were incubated overnight with TEV protease (1:40 mass ratio) and loaded onto gel filtration using a superdex 200 16/60 column equilibrated in 50 mM HEPES, 500 mM NaCl, 0.5 mM TCEP. Both samples were found to crystallize with the majority of the crystals coming from the SP flow through which had greater yield although slightly less pure.

**Crystallization.** All crystals were grown from and optimized using pre-prepared mixes from the Morpheus screen from molecular dimensions. Phosphate bound crystals (used for fragment screening) were grown at 20 degrees from conditions containing 20 % Ethylene Glycol, 10 % PEG 8 K, 0.05 M HEPES pH, 0.05 M MOPS, 0.03 M Sodium Nitrate, 0.03 M Sodium Phosphate, 0.03 M Ammonium Sulphate using 10 mg/ml protein. For crystal optimization seeding was performed: 5–10 crystals were crushed with glass probe and transferred to 25 μl of well solution. A seed bead was added, and the mixture was sonicated for around 30–60 s with pulsing. Final seeding was performed with a 1 in 400 dilutions of seed stock. Final plates were set up with protein at 5 mg/ml (diluted fourfold in water from 20 mg/ml stock) with a slightly reduced precipitant concentration (16% Ethylene Glycol, 8% PEG 8 K, 0.05 M HEPES, 0.05 M MOPS, 0.03 M Sodium Nitrate, 0,03 M Sodium Phosphate, 0.03 M Ammonium Sulphate), using 300 nl drops (1:1 ratio) with 20 nl seeds (added last). Crystals without phosphate were grown at 10 mg/ml from conditions containing 20% Ethylene Glycol, 10% PEG 8 K, 0.05 M MES pH 6.5, 0.05 M Imidazole pH 6.5, 10% v/v Ethylene glycols mix (contains 0.3 M Diethylene glycol, 0.3 M Triethylene glycol, 0.3 M Tetraethylene glycol, 0.3 M Pentaethylene glycol). For crystals containing AMP-PNP, 10 mM AMP-PNP and 10

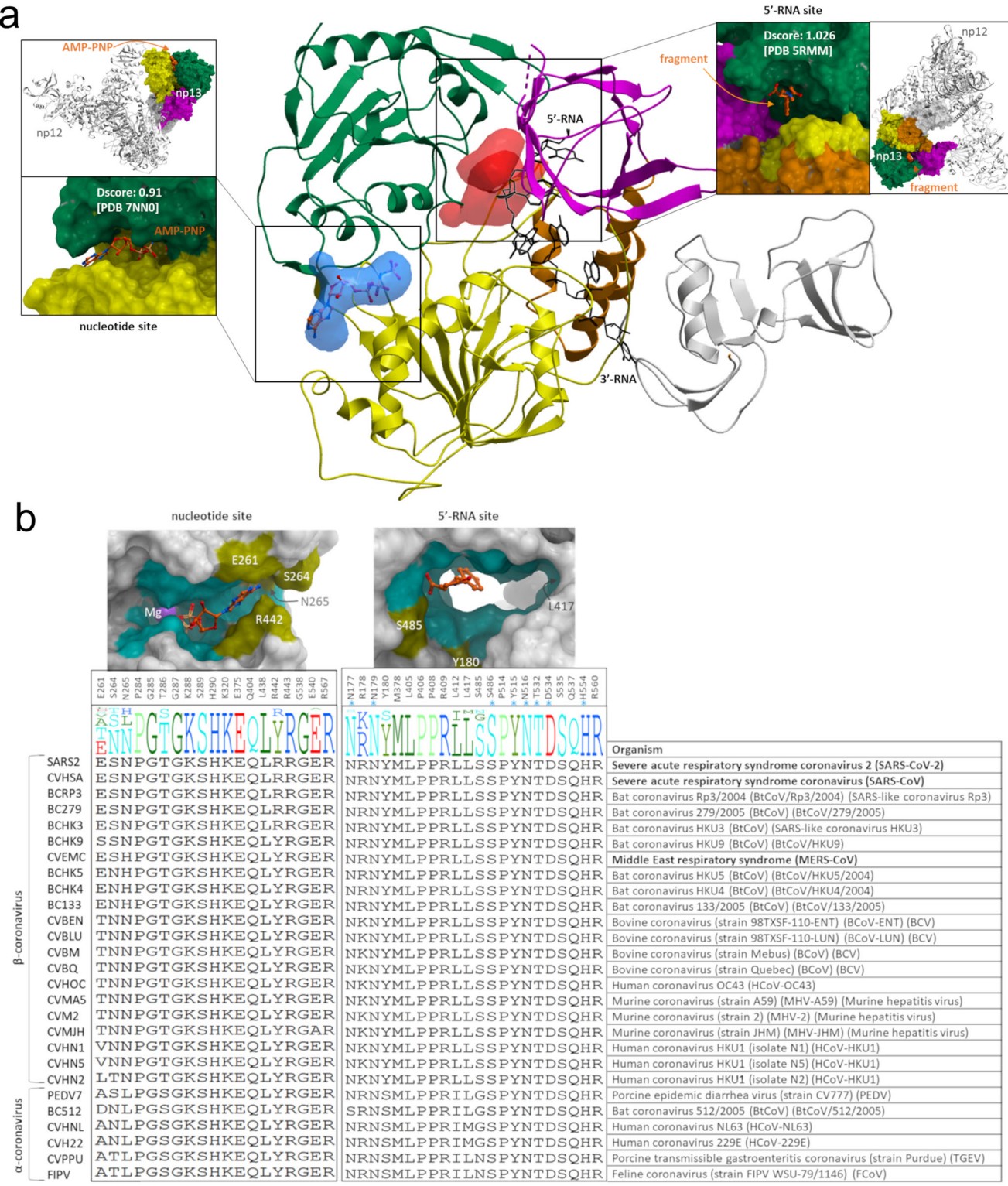

**Fig. 6 Druggability and genetic variability of the nucleotide and 5′-RNA sites. a** The nucleotide binding pocket (blue) occupied by AMP-PNP and a cavity (red) that is partially occupied by the 5′-end of the substrate RNA in the transcription complex structure (PDB 7CXM) were identified as druggable sites in our crystal structures. Druggability scores (Dscores) calculated with SiteMap[33] are indicated (insets). The location of each binding site in the context of the transcription complex structure is also shown (insets). **b** Conservation of sidechains lining the AMP-PNP-bound site (left) and the fragment-bound 5′-RNA site (right) is shown across 27 α- and β-coronavirus sequences reviewed in the Uniprot database. Conserved (cyan) and non-conserved (mustard) sidechains are mapped on 3D structures. Asterisks indicate sidechains that are in direct contact with the fragment in structure 5RMM. Residue numbering in SARS-CoV-2 is shown at the top. Viruses associated with human epidemics are highlighted in bold.

mM $MgCl_2$ were added to the protein and incubated on ice for 10 minutes. Crystallization was performed at 5 mg/ml and crystals appeared at 20° in conditions containing 20% Ethylene Glycol, 10% PEG 8 K, 0.05 M MES pH 6.5, 0.05 M Imidazole pH 6.5, 10% v/v Alcohols mix (contains 0.2 M 1,6-Hexanediol, 0.2 M 1-Butanol, 0.2 M 1,2-Propanediol, 0.2 M 2-Propanol, 0.2 M 1,4-Butanediol, 0.2 M 1,3-Propanediol). All crystals were loop mounted and flash cooled in liquid nitrogen without the addition of further cryoprotectant.

**Structure determination.** All data were collected at Diamond light source beamline I04-1 and processed using XDS[37] and DIALS[38]. The structures were solved by molecular replacement using the program PHASER and the structure of SARS-CoV-1 NSP13 (6JYT) as a search model. Refinement was performed using PHENIX REFINE[39]. A summary of the data collection and refinement statistics are shown in Table 1.

**X-ray fragment screening.** A total of 648 fragments from the DSI poised and York3D libraries (500 mM stock concentration dissolved in DMSO) were transferred directly to NSP13 crystallization drops using an ECHO liquid handler (50 mM nominal final concentration), and soaked for 1–3 h before being loop mounted and flash cooled in liquid nitrogen. A total of 616 datasets were collected at a resolution of 2.8 Å or higher with the majority being in the range of 1.8 Å to 2.4 Å. Data were collected at Diamond light source beamline I04-1 and processed using the automated XChem Explorer pipeline. Structures were solved by difference Fourier synthesis using the XChem Explorer pipeline[40]. Fragment hits were identified using the PanDDA[32] program. Refinement was performed using REFMAC[41] or BUSTER. A summary of data collection and refinement statistics for all fragment bound datasets is shown in Supplementary Data 1.

**Druggability and conservation analysis.** The PocketFinder function implemented in ICM (version 3.9-2b) (Molsoft, San Diego) was used to map binding pockets on our apo and AMP-PNP complex structures of NSP13[42]. Druggability was calculated with SiteMap (Release 2019-4 of Maestro—Schrodinger, New-York)[33]. The AMP-PNP complex structure was used to calculate the druggability of the nucleotide site after removing the bound nucleotide and its coordinating magnesium. The druggability of the 5′-RNA site was calculated on the fragment-bound complex structure (PDB 5RMM) after removing the fragment. The multiple alignments of twenty-seven coronavirus helicase sequences was carried out with ICM[43].

**Reporting summary.** Further information on research design is available in the Nature Research Reporting Summary linked to this article.

## Data availability

Crystallographic coordinates and structure factors for all structures have been deposited in the Protein Data Bank with the following accession codes: 6ZSL, 7NIO, 7NN0, 5RL6, 5RL7, 5RL8, 5RL9, 5RLB, 5RLC, 5RLD, 5RLE, 5RLF, 5RLG, 5RLH, 5RLI, 5RLJ, 5RLK, 5RLL, 5RLM, 5RLN, 5RLO, 5RLP, 5RLQ, 5RLR, 5RLS, 5RLT, 5RLU, 5RLV, 5RLW, 5RLY, 5RLZ, 5RM0, 5RM1, 5RM2, 5RM3, 5RM4, 5PM5, 5RM6, 5RM7, 5RM8, 5RM9, 5RMA, 5RMB, 5RMC, 5RMD, 5RME, 5RMF, 5RMG, 5RMH, 5RMI, 5RMJ, 5RMK, 5RML, 5RMM, 7NNG. Ground state datasets and ground state model used for the PanDDA analysis have been deposited as a multi dataset entry under the accession code: 5ROB.

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

## Acknowledgements

The crystallographic screen was supported by the XChem facility at Diamond Light Source (proposal ID LB26998). We thank all the staff of Diamond Light Source for providing support and encouragement which allowed us to carry out this work during the COVID-19 lockdown. The SGC is a registered charity (number 1097737) that receives funds from AbbVie, Bayer Pharma AG, Boehringer Ingelheim, Canada Foundation for Innovation, Eshelman Institute for Innovation, Genome Canada, Innovative Medicines Initiative (EU/EFPIA) [ULTRA-DD grant no. 115766], Janssen, Merck KGaA Darmstadt Germany, MSD, Novartis Pharma AG, Ontario Ministry of Economic Development and Innovation, Pfizer, São Paulo Research Foundation-FAPESP, Takeda, and Wellcome [106169/ZZ14/Z].

## Author contributions

O.G., F.V.D., and J.A.N. initiated the project. Y.Y. and J.A.N. performed expression, protein purification, and crystallization. A.D. and J.A.N. performed crystal optimization. A.D., A.A., J.B.-N., D.F., and L.D. Fragment soaking, Crystal mounting, and XChem data management. J.A.N. A.D., A.A., J.B.-N., D.F., L.D., T.G.-S., and R.S. performed X-ray data analysis and review of fragment binding. S.Y. and M.S. performed druggability and conservation analysis. J.A.N., O.G., S.Y., and M.S. wrote the original draft manuscript. All authors read and approved the manuscript.

## Competing interests

The authors declare no competing interests.
