## [Peer Review File · Nature Communications]

Reviewers' Comments:

Reviewer #1:

Remarks to the Author:

Newman and colleagues report crystallographic investigations of SARS-CoV-2 nsp13 helicase. Although SARS-CoV-2 nsp13 is only different from SARS-CoV-1 nsp13 (crystal structure determined by Jia et al in 2019) by a single amino acid substitution, the structures determined by the authors of this paper has significantly higher resolution. Some structure exceeds 2.0Å. High resolution structures are important to structure-based drug development. The authors established robust crystallization of nsp13 protein and carried out crystallographic fragment-based screening; over 600 crystals were soaked, and they identified 65 fragments bound to nsp13 from 52 datasets; fragments probed multiple sub-pockets within NTPase active site, RNA binding channel, ZBD as well as some allosteric sites. From which, they identified two druggable sites for further drug development.

Main points

1. The authors obtained different nsp13 structures complexed by AMP-PNP, Mg⁺⁺ or phosphate, which represent different state of ATP hydrolysis and DNA translocation. Comparing these structures, the authors observed evident interdomain movements and they proposed a model for the translocation mechanism of nsp13. This inchworm model is overall in line with the helicase mechanism of other SF1B helicases. Recently, two papers studying nsp13 unwinding mechanism using single molecule methods were published (Biophys J . 2021 Mar 16;120(6):1020-1030. doi: 10.1016/j.bpj.2020.11.2276; Mechanism of duplex unwinding by coronavirus nsp13 helicases bioRxiv doi: <https://doi.org/10.1101/2020.08.02.233510>). Could the authors compare their model with results from those investigations in the discussion section?
2. The authors did an amazing work to establish robust crystallization of nsp13 helicase essential for fragment-based screening (FBS). Nsp13 is among the most conserved proteins in CoVs and Nidoviruses; thus, it is a potential wide-spectrum drug target. Concerning FBS method, has the authors tried covalent fragment library? given nsp13 has many cysteines on ZBD domain. ZBD is important to the interaction of nsp13 with other protein binding partners and the modulation of nsp13 activity. Fragments with nucleophilic warheads may also lead to novel drug leads.
3. Fragments identified by crystal soaking might not bind the protein tightly. However, preliminary assessment of fragment-nsp13 binding may help to improve the hit rate. Have the authors used some methods to evaluate binding affinity of the fragments?
4. Is there any chance, some of those fragments exhibit detectable inhibitory effects on NTPase activity of helicase activity of nsp13?

Specific points

1. The Introduction section should include some structural comparison of different CoV nsp13 helicases with other Nidovirus nsp10 helicases, such as EAV nsp10 and PRRSV nsp10 helicases, given viral helicase is one of the most conserved proteins in Nidovirales.
2. Could the author explain when AMP-PNP is bound to
3. K320 recognizes 2'-OH of the ribose, suggesting nsp13 prefers ribonucleotides. Nsp13 lacks nucleotide specificity, it can even hydrolyze dNTP. Could the authors explain this contradiction?
4. I guess there is a typo in this sentence: "... such that the respective interfaces are maintained whilst the 1B domain pivots around its relatively narrow point of attachment at the terminus of the stalk domain ...". More details should be present here to fully understand the conformation variation of the 1B domain.
5. "We have determined the crystal structures of full-length SARS-CoV-2 NSP13 (residues 1-901)", I think the number of residues comprising nsp13 is around 600.
6. CC 1/2 of AMP-PNP complex structure at high resolution shell is 0.359, this is quite low and may reflect poor data quality in this range.
7. Could the author provide an outlook of the next step of drug design based on a wealth of structural

data of fragment-nsp13 obtained in current study.

Reviewer #2:

Remarks to the Author:

This manuscript describes crystal structures of the SARS-CoV-2 helicase in the apo-form and in the presence of both phosphate and AMP-PNP. Interestingly, in the nucleotide bound form, AMP-PNP is bound to all 4 molecules in an ASU but in two distinct modes. Most importantly, the authors carried out the fragment screening which identified hot spots including the nucleotide and nucleic acid binding sites. Although viral helicases have long been considered to be good targets for development of antivirals, it is technically difficult to discover lead compounds specifically targeting them due to the large conformational changes in their hydrolysis cycles. In addition, compounds targeting the nucleotide binding site in the viral helicases usually cause concerns that the inhibitors may interfere with activity of the cellular helicases as well. As such, the data presented in this manuscript are exciting, which should appeal to the readership of Nature Communications.

Major concerns:

1. The authors need to show the arrangement of the molecules of AMP-PNP bound helicase in an ASU and clarify whether the different binding modes are due to the crystal packing.
2. The authors need to tell if SARS-CoV-2 helicase has nucleotide specificity or not from structural analysis.
3. Please show the bound fragments both in a sigma-sA-weighted 2mFo-DFc electron density map and a simulated annealing 2mFo-DFc omit map in the supplementary figures.

Minor concerns:

1. Page 1, Introduction section, "outbreaks of SARS-CoV-1 in 2002(4) and MERS in 2012(5)". SARS-CoV-1 should be SARS (the causing virus is called SARS-CoV).

Response to reviewer comments

We wish to thank both reviewers for their careful consideration of our manuscript and generally supportive comments. We have provided a point by point response to the specific points made in the text below.

REVIEWER COMMENTS

Reviewer #1 (Remarks to the Author):

Newman and colleagues report crystallographic investigations of SARS-CoV-2 nsp13 helicase. Although SARS-CoV-2 nsp13 is only different from SARS-CoV-1 nsp13 (crystal structure determined by Jia et al in 2019) by a single amino acid substitution, the structures determined by the authors of this paper has significantly higher resolution. Some structure exceeds 2.0Å. High resolution structures are important to structure-based drug development. The authors established robust crystallization of nsp13 protein and carried out crystallographic fragment-based screening; over 600 crystals were soaked, and they identified 65 fragments bound to nsp13 from 52 datasets; fragments probed multiple sub-pockets within NTPase active site, RNA binding channel, ZBD as well as some allosteric sites. From which, they identified two druggable sites for further drug development.

Main points

1. The authors obtained different nsp13 structures complexed by AMP-PNP, Mg⁺⁺ or phosphate, which represent different state of ATP hydrolysis and DNA translocation. Comparing these structures, the authors observed evident interdomain movements and they proposed a model for the translocation mechanism of nsp13. This inchworm model is overall in line with the helicase mechanism of other SF1B helicases. Recently, two papers studying nsp13 unwinding mechanism using single molecule methods were published (Biophys J . 2021 Mar 16;120(6):1020-1030. doi: 10.1016/j.bpj.2020.11.2276; Mechanism of duplex unwinding by coronavirus nsp13 helicases bioRxiv doi: <https://doi.org/10.1101/2020.08.02.233510>). Could the authors compare their model with results from those investigations in the discussion section?

This is a good suggestion as these two studies do probe the mechanism of NSP13 with a different methodology. At first appearance it may seem like some of the data in the Hu *et al* preprint is at odds with our mechanism as according to their data the unwinding reaction appears to proceed in steps of several base pairs (4-9 bases depending on nucleotide) with possibility of additional hydrolysis events building up tension in a “spring loaded mechanism”. Our molecular mechanism is based on a single translocation per nucleotide hydrolysis cycle, although we did make the distinction (partly because of the data by Mickolajczyk *et al*) that this is a mechanism for directional NSP13 translocation along single stranded DNA/RNA, and not an unwinding mechanism. We suggest that the lack of a wedge or hairpin like feature (as found on other helicases) to aid the strand separation may explain why in the unwinding reaction the enzyme appears to proceed several bases at a time. Either the spring loaded type mechanism suggested by Hu *et al* or a more general passive mechanism (helicase opportunistically advances upon the spontaneous opening of base pairs) would presumably be significantly stimulated by destabilizing forces on the DNA as shown in the recent Mickolajczyk *et al* paper. The mutagenesis data in

the Hu *et al*/ paper is also in good agreement with our molecular model and whilst the suggested mechanism for the accumulation of tension by means of a flexible 1B and Stalk domain is not a feature of our model we do see significant flexibility in the position adopted by this region in our structures.

We have altered the discussion section to illustrate these points.

“We do not describe an active base separating mechanism for NSP13 consistent with recent biochemical and single molecule studies on NSP13. In the RNA unwinding reaction NSP13 was a predominantly passive helicase (advancing upon the spontaneous opening of base pairs), with a peak step size of 2 base pairs (interpreted to be 2 rate limiting ATP binding events) and a strong force dependent stimulation of activity that suggests mechanoregulation by the RNA polymerase NSP12(8). Single molecule FRET studies of DNA unwinding by NSP13 show larger step sizes of up to 4-9 base pairs depending on nucleotide used which the authors suggest may indicate a “spring loaded” unwinding mechanism with the accumulation of tension achieved by means of flexibility in the 1B and stalk domains(30). Whilst the larger step sizes of the unwinding mechanisms contrast with our single step translocation mechanism, all models feature the same stoichiometry of a single nucleotide hydrolysis event per base. We suggest that the lack of an active strand separating hairpin or wedge to aid strand separation NSP13 is unable to translocate with the same efficiency and must presumably pause to either accumulate tension or wait for the spontaneous opening of DNA/RNA.”

2. The authors did an amazing work to establish robust crystallization of nsp13 helicase essential for fragment-based screening (FBS). Nsp13 is among the most conserved proteins in CoVs and Nidoviruses; thus, it is a potential wide-spectrum drug target. Concerning FBS method, has the authors tried covalent fragment library? given nsp13 has many cysteines on ZBD domain. ZBD is important to the interaction of nsp13 with other protein binding partners and the modulation of nsp13 activity. Fragments with nucleophilic warheads may also lead to novel drug leads.

This is a good suggestion although we did not test any covalent fragments against NSP13. As the reviewer points out the zinc binding domain is cysteine rich and includes 3 zinc fingers that could be targeted by fragments containing covalent warheads. Indeed, zinc ejection from Zn fingers is a possible means of inhibition that is being tested for targeting of the nucleocapsid protein 7 from HIV virus. We note that whilst the covalent fragment screen for the SARS-CoV-2 Mpro protease revealed many promising covalent inhibitor series, these have now largely been de-prioritized for further development into antiviral therapeutics due to safety concerns. We assume the same concerns would apply to covalent compounds targeting NSP13 (perhaps even more so given the likely greater reactivity of the Mpro catalytic cysteine) and for this reason we have focussed our initial efforts on non-covalent fragments. Our structures and robust crystallization conditions do however provide a platform for such screening to be performed.

3. Fragments identified by crystal soaking might not bind the protein tightly. However, preliminary assessment of fragment-nsp13 binding may help to improve the hit rate. Have the authors used some methods to evaluate binding affinity of the fragments?

This is a good point that should probably be made more explicit in the manuscript. Given that in our screen we tested around 650 compounds with molecular weights around 300 Daltons it would be quite

unrealistic to expect any of the hits to have particularly potent binding (based on our experience with other projects crystallographic fragments range from unmeasurable to a maximum of around 100 μM level). We have not yet tested the affinity of the directly observed crystallographic fragments although this will be performed on selected fragments as part of our follow up strategy. We have added the following to the summary section to make this clearer.

“Whilst these fragments do represent useful starting points for inhibitor development they have not yet been validated as inhibitors in biochemical assays or as binders in alternate biophysical assays and are not expected to be potent inhibitors without further optimization.”

4. Is there any chance, some of those fragments exhibit detectable inhibitory effects on NTPase activity of helicase activity of nsp13?

The fragments due to their small size and the entropic penalties associated with binding tend to bind with good ligand efficiencies and make relatively high-quality interactions with the protein that are often maintained in potent lead or drug molecules. Our hope is that follow up molecules based on these fragments can deliver potency either by fragment growing or merging. Molecules based on our fragments identified do bind to sites that would be expected to inhibit the enzyme (Nucleotide and RNA competitive binders) and thus we would expect inhibition to closely match with potency for these classes of fragments.

Specific points

1. The Introduction section should include some structural comparison of different CoV nsp13 helicases with other Nidovirus nsp10 helicases, such as EAV nsp10 and PRRSV nsp10 helicases, given viral helicase is one of the most conserved proteins in Nidovirales.

We have now included a wider comparison to other Nidovirus helicases in the introduction section. The new added section on page 2 contains.

“This same basic 5-domain architecture is shared by other Nidovirus helicases such as the NSP10 proteins from Equine arteritis virus (15) and Porcine reproductive and respiratory syndrome virus (24) and to a lesser extent the human nonsense mediated mRNA decay factor UPF1(17), which feature a structurally similar helicase core (R.M.S.D around 3.0\AA) and more diversity in terms of composition of the zinc binding domain and connection between the 1B domain and helicase core.”

2. Could the author explain when AMP-PNP is bound to

3. K320 recognizes 2'-OH of the ribose, suggesting nsp13 prefers ribonucleotides. Nsp13 lacks nucleotide specificity, it can even hydrolyze dNTP. Could the authors explain this contradiction?

This is a good point, we do see a contact between K320 and the 2'OH in the mode B AMP-PNP complex although from our structures this would suggest a preference for ribonucleotides rather than an inability to hydrolyse deoxynucleotides. A recent preprint by Hu *et al* does indicate that whilst NSP13 is able to hydrolyse a variety of NTP's there are preferences with significantly greater turnover rates for ATP

compared to other Nucleotide triphosphates or deoxynucleotide triphosphates. We have included the following in the manuscript in the results section to make this point.

“In our structures, the majority of the contacts made around the adenine and ribose moieties do not appear to be specific to a single nucleotide, consistent with the observation that NSP13 is able to hydrolyze multiple nucleotides including deoxy-ribonucleotides in enzymatic assays(14,30). A single contact between the adenine N6 to the main chain carbonyl of E261, and the ribose 2' OH and K320 indicates a possible preference for both adenine and ribonucleotides, consistent with a recent single molecule analysis of NSP13 *in vitro*(30).”

4. I guess there is a typo in this sentence: “... such that the respective interfaces are maintained whilst the 1B domain pivots around its relatively narrow point of attachment at the terminus of the stalk domain ...” More details should be present here to fully understand the conformation variation of the 1B domain.

This was not actually a typo but we can see how this sentence may be difficult to understand. The point we were trying to make was that the movement of the 1B domain appears to be caused by the opening and closing of the 1A and 2A domains with the 1B domain pivoting around its connection with the stalk domain. In both the open and closed states interface between the 1B domain and the helicase core which is based on contacts made to loop 337-340 in the 1A domain and loops 483-487 and 513-517 in the 2A domain, are relatively unchanged.

We have rephrased this section to make it clearer.

“In this case the movements do correlate with the relative positioning of the 1A and 2A domains, and the 1B domain can be seen to pivot around its narrow point of connection to the stalk domain. In both the open and closed states, the 1B domain forms the same basic interface to the helicase core, forming contacts to loop 337-340 in the 1A domain and loops 483-487 and 513-517 in the 2A domain (Figure 2B).

5. “We have determined the crystal structures of full-length SARS-CoV-2 NSP13 (residues 1-901)”, I think the number of residues comprising nsp13 is around 600.

This has now been corrected it should be residues 1-601.

6. CC 1/2 of AMP-PNP complex structure at high resolution shell is 0.359, this is quite low and may reflect poor data quality in this range.

We don't really disagree that the data is quite weak in this resolution range, however judging by other metrics such as I/σ (1.2) the data quality in the highest resolution shell for this data set still likely contain useful information. In line with recent best practices in structure refinement, and given the fact that the structure has already been released and electron density maps are of otherwise good quality we don't think that truncating the data would achieve any useful purpose at this stage and we have decided not to change anything.

7. Could the author provide an outlook of the next step of drug design based on a wealth of structural data of fragment-nsp13 obtained in current study.

Direct X-ray fragment screening is still a relatively new technique and a consensus on how to progress fragments has not yet been reached. The fragments bound to NSP13 are located in multiple pockets with multiple means of inhibition possible (ATP competitive, RNA competitive or allosteric), and at this early stage we suggest maintaining diversity in terms of pockets, although our evolutionary and druggability analysis would suggest the nucleotide and 5'-RNA binding pockets as highly druggable and well conserved evolutionarily. The fragments are generally well represented in commercially available compound collections and computational tools such as docking and hotspot mapping may be useful in guiding the choice of follow up molecules. The fragalysis website hosted by Diamond light source (<https://fragalysis.diamond.ac.uk>) contains the collection NSP13 fragments pre-aligned and categorized by binding site, with tools to select follow up molecules based on querying chemical databases, and can aid in visualizing possibilities for either growing or linking fragments in order to progress the potency

Reviewer #2 (Remarks to the Author):

This manuscript describes crystal structures of the SARS-CoV-2 helicase in the apo-form and in the presence of both phosphate and AMP-PNP. Interestingly, in the nucleotide bound form, AMP-PNP is bound to all 4 molecules in an ASU but in two distinct modes. Most importantly, the authors carried out the fragment screening which identified hot spots including the nucleotide and nucleic acid binding sites. Although viral helicases have long been considered to be good targets for development of antivirals, it is technically difficult to discover lead compounds specifically targeting them due to the large conformational changes in their hydrolysis cycles. In addition, compounds targeting the nucleotide binding site in the viral helicases usually cause concerns that the inhibitors may interfere with activity of the cellular helicases as well. As such, the data presented in this manuscript are exciting, which should appeal to the readership of Nature Communications.

Major concerns:

1. The authors need to show the arrangement of the molecules of AMP-PNP bound helicase in an ASU and clarify whether the different binding modes are due to the crystal packing.

This is a valid point given the observation of significant flexibility in NSP13 structures, we have included an additional supplementary figure showing the arrangement of molecules in the AMP-PNP crystallographic asymmetric unit. Given all the chains contain AMP-PNP bound yet only one chain adopts the closed conformation the influence of crystal contacts can not be ruled out. We do believe that these structures are representative states for NSP13 in solution and support for this can be observed in the recent cryo EM structures of NSP13 in complex with the replication and transcription complex (reference 19) which contains two NSP13 molecules that show conformations equivalent to the open and closed states in our study.

2. The authors need to tell if SARS-CoV-2 helicase has nucleotide specificity or not from structural analysis.

This is a good point that was also raised by reviewer 1. From our structures the adenine moiety is positioned between H290 and R442 we do not see contacts around that would ensure recognition of

ATP specifically (such as those provided by the Q motif or motif0 in some SF2 helicases). There is a single contact from the adenine N6 to the main chain carbonyl of E261 in the pre hydrolysis state structure and a single contact to between K320 and the AMP-PNP 2'OH in the mode B complex which we suggest may indicate a slight preference (rather than a requirement) for ATP and ribonucleotides respectively. We have added the following to the results section.

“In our structures, the majority of the contacts made around the adenine and ribose moieties do not appear to be specific to a single nucleotide, consistent with the observation that NSP13 is able to hydrolyze multiple nucleotides including deoxy-ribonucleotides in enzymatic assays(14,30). A single contact between the adenine N6 to the main chain carbonyl of E261, and the ribose 2' OH and K320 indicates a possible preference for both adenine and ribonucleotides, consistent with preferences identified in a recent single molecule analysis of NSP13 in vitro(30).”

3. Please show the bound fragments both in a sigma-sA-weighted 2mFo-DFc electron density map and a simulated annealing 2mFo-DFc omit map in the supplementary figures.

We are happy to include a figure with the electron density maps in the supplementary information, after all the primary evidence for the binding of the fragments is the electron density maps. Due to the ability of the PANDDA algorithm to identify hits in electron density maps below conventional electron density thresholds we have also included the PANDDA event maps in the figure. We understand the request for simulated annealing omit maps (for removal of model bias) but for primarily technical reasons we have included instead omit maps calculated by DIMPLE as part of the PANDDA pipeline using the ground state model. These maps are calculated before any ligand modelling has taken place and thus are bias free and were used due to the significant computational resources required for the simulated annealing procedure (run time of around 10 hours per dataset using the PHENIX map tool). As can be seen in table S3, most of the NSP13 fragment hits show clear electron densities in conventional difference maps although a minority, can only be visualized at lower contour electron density contour levels and would likely not have been identified if not for using PANDDA. We have decided to contour the omit and 2mFo-DFc maps as a constant 1σ level so as to enable comparison of electron density levels between datasets.

Minor concerns:

1. Page 1, Introduction section, "outbreaks of SARS-CoV-1 in 2002(4) and MERS in 2012(5)". SARS-CoV-1 should be SARS (the causing virus is called SARS-CoV).

This has now been changed

Reviewers' Comments:

Reviewer #1:

Remarks to the Author:

I believe the authors have addressed all questions, and I am satisfied with the answers. The paper is in current form is acceptable for publication.